# Caesarean section delivery and childhood obesity in a British longitudinal cohort study

Gwinyai Masukume[1,2], Ali S. Khashan[1,3], Susan M. B. Morton[4], Philip N. Baker[5], Louise C. Kenny[6], Fergus P. McCarthy[1,2,7] *

**1** INFANT Research Centre, Cork, Ireland, **2** Department of Obstetrics and Gynaecology, University College Cork, Cork, Ireland, **3** School of Public Health, Western Gateway Building, University College Cork, Cork, Ireland, **4** Centre for Longitudinal Research–He Ara ki Mua, University of Auckland, Auckland, New Zealand, **5** College of Life Sciences, University of Leicester, Leicester, England, United Kingdom, **6** Department of Women's and Children's Health, Institute of Translational Medicine, Faculty of Health and Life Sciences, University of Liverpool, Liverpool, England, United Kingdom, **7** Department of Women and Children's Health, School of Life Course Sciences, King's College London, London, England, United Kingdom

* Fergus.mccarthy@ucc.ie

## Abstract

### Background

Several studies reported an association between Caesarean section (CS) birth and childhood obesity. However, there are several limitations in the current literature. These include an inability to distinguish between planned and emergency CS, small study sample sizes and not adjusting for pre-pregnancy body-mass-index (BMI). We examined the association between CS delivery and childhood obesity using the United Kingdom Millennium Cohort Study (MCS).

### Methods

Mother-infant pairs were recruited into the MCS. Use of sampling weights ensured the sample was representative of the population. The exposure was categorised as normal vaginal delivery (VD) [reference], assisted VD, planned CS and emergency CS. Childhood obesity prevalence, at age three, five, seven, eleven and fourteen years was calculated using the International Obesity Taskforce criteria. Mixed-effects linear regression models were fitted with associations adjusted for several potential confounders like maternal age, pre-pregnancy BMI, education and infant macrosomia. Linear regression models were fitted evaluating body fat percentage (BF%), at age seven and fourteen years.

### Results

Of the 18,116 infants, 3872 (21.4%) were delivered by CS; 9.2% by planned CS. Obesity prevalence was 5.4%, 5.7%, 6.5%, 7.1% and 7.6% at age three, five, seven, eleven and fourteen years respectively. The mixed-effects linear regression model showed no association between planned (adjusted mean difference = 0.00; [95% confidence interval (CI) -0.10; 0.10], p-value = 0.97) or emergency CS (adjusted mean difference = 0.08; [95% CI -0.01; 0.17], p-value = 0.09) and child BMI. At age seven years, there was no association between

**Data Availability Statement:** The authors collected data from the United Kingdom Data Service https://www.ukdataservice.ac.uk/. The Millennium Cohort Study data is available free of charge to interested

researchers. Authors confirm that these datasets can be accessed by others in the same manner as the authors and they did not have special access privileges.

**Funding:** G.M. is supported by the Irish Centre for Fetal and Neonatal Translational Research (INFANT) (grant no. 12/RC/2272). The funders had no role in study design, data collection and analysis, decision to publish, or preparation of the manuscript.

**Competing interests:** The authors have declared that no competing interests exist.

planned CS and BF% (adjusted mean difference = 0.13; [95% CI -0.23; 0.49]); there was no association at age fourteen years.

## Conclusions

Infants born by planned CS did not have a significantly higher BMI or BF% compared to those born by normal VD. This may suggest that the association, described in the literature, could be due to the indications/reasons for CS birth or residual confounding.

## Introduction

As summarised by several systematic reviews and meta-analyses[1–5], numerous studies have found a consistent association between Caesarean section (CS) birth and subsequent childhood obesity. However, it remains unclear if this association indicates that CS causes obesity in childhood or is indicative of underlying confounding factors. A trial randomising pregnant women to deliver by CS or vaginally (VD) would provide definitive evidence.[6] In the absence of this clinical trial, data from observational studies, albeit limited by the paucity and small sample size of relevant studies, have been leveraged by controlling for major confounding variables, notably from maternal pre-pregnancy body mass index (BMI),[7] by considering obesity in siblings discordant for birth mode,[8, 9] and by comparing those born by elective and emergency CS.[10–14] Animal[15, 16] and microbial studies[17, 18] have also helped to investigate this question.

Differences in the infant gut microflora, which influence nutrient uptake, is the main hypothesised mechanism by which childhood obesity develops following CS delivery in offspring.[19–21] Differential exposure to the vaginal, perineal and faecal microflora between infants born by CS, particularly elective CS, and those born vaginally is presumed to determine the initial composition of an infant's gut microflora.[22, 23] There is the contentious possibility, however, that the putative placental microbiota influences composition too, regardless of delivery mode.[24, 25] Another potential mechanism relates to differences between infants born by CS and VD in the intrapartum concentration of cortisol, noradrenaline and other inflammatory chemicals,[26, 27] which may result in long term neuro-immuno-endocrine, epigenetic and other changes which may influence energy metabolism.

Studying the associations underlying the role of CS with childhood obesity is important, given the global increase in CS rates and the epidemic of childhood obesity.[28–30] We recently performed two studies[10, 31] to address some of the limitations of previous reports, but both studies only followed-up offspring to age five years.

According to the systematic reviews and meta-analyses estimates of the strength of association between birth mode and childhood obesity, albeit with bias favouring positive effects, have been generally less than a relative risk of 1.50.[3, 4]

We aimed to investigate the association between planned/elective CS, a potentially modifiable risk factor, and childhood obesity using a large contemporary prospective longitudinal cohort study. In this study we used a similar approach to our previous work but with a different and larger dataset and much longer follow-up. This included analysis of the link between CS birth and body fat percentage (BF%) as previously performed,[31] on the basis that adiposity may be a more accurate measure of obesity than BMI.[32]

## Materials and methods

The Millennium Cohort Study (MCS) is an ongoing multidisciplinary nationally representative longitudinal cohort study. At approximately nine months of age, children born in the

United Kingdom (UK) from September 2000 through to January 2002 were recruited into the study, with over-sanpling for ethnic minorities. The overall sample was representative of the population. A total of 18,827 infants were enrolled. To date there have been six major data collection sweeps at nine months, three, five, seven, eleven and fourteen years of age. Data was collected by trained interviewers using validated procedures and instruments. Further comprehensive details about the MCS are available from its cohort profile [33]. Ethical approval for the Millennium Cohort Study surveys was granted by the London Multicentre Research Ethics Committee.

The exposure, mode of birth, was classified as normal or assisted VD and planned or emergency CS. Assisted VD constituted birth by forceps or vacuum extraction. Planned and emergency CS were mainly pre-labour or in labour respectively.[10]

Height was measured using a Leicester height measure. Weight and BF % were measured using Tanita[TM] scales; the latter was ascertained by the scale's bioelectric impedance mechanism. BMI in kg/m$^2$ was classified as thin, normal, overweight or obese according to the standard International Obesity Task Force (IOTF) criteria, which are sex and age specific.[34–36]. Of the major BMI classification systems, including those from the World Health Organization (WHO) and Centers for Disease Control and Prevention (CDC), the IOTF criteria have been the most frequently used for this research topic.[3, 37] Using the 2006 WHO child growth standards, anthropometric z-scores were also calculated.[38]

## Statistical analysis

Stata version 14SE (StataCorp LP College Station, TX) was used for statistical analysis. Categorical variables were described using frequencies (n) and percentages (%). Numeric variables were described using the mean (standard deviation-SD) or median (interquartile range-IQR). In the main analysis, to account for the continuous BMI, repeated measures available at age three, five, seven, eleven and fourteen years, crude and adjusted mixed-effects linear regression models were generated. In secondary analysis, to replicate our prior work,[10] multinomial logistic regression models were fitted to investigate the association between birth mode and IOTF BMI category transition between age three and five years; 0 = remained normal (base outcome), 1 = remained obese, 2 = became obese, 3 = became non-obese and 4 = any other transition. Linear regression models were fitted to investigate the association between birth mode and BF%, available at age seven and fourteen years.

Based on prior literature, potential confounders were defined *a priori*. These included maternal age, ethnicity, education, marital status, couple income, infant sex, birth weight, smoking during pregnancy, gestational age, diabetes mellitus, parity, and pre-pregnancy BMI. We and other researchers found that infant macrosomia explained significant associations,[10, 31] we thus considered it as a potential confounder. Sub-group analysis was performed for infants with mothers aged > 35 years, born pre-term (< 37 weeks) and by their sex. A p-value < 0.05 was considered to be statistically significant.

## Missing data

Multiple imputation was performed for maternal pre-pregnancy BMI and childhood BF% which all had substantial amounts of missing data. We assumed this data to be missing at random.[39] Variables in the main analysis were included in the imputation model. Forty-five imputations were done and the results were pooled according to Rubin's rules.[40] Imputed values were checked for plausibility in relation to observed values.

## Results

The final baseline population consisted of 18,116 (96.2%) mother-infant pairs following exclusion of infants with an unknown mode of delivery (143, 0.76%), multiple births (467, 2.48%) and where the main respondent was not the infant's biologic mother because some potentially confounding variables were available only where mothers were the respondents.

Of the 18,116 infants, 3872 (21.4%) were delivered by CS; planned CS (9.2%), emergency CS (12.2%), normal VD 12,567 (69.4%) and assisted VD 1,677 (9.3%) (Table 1). At birth, 10.8% of the infants were macrosomic (> 4kg). The IOTF prevalence of obesity at ages three, five, seven, eleven and fourteen years of age was 5.4%, 5.7%, 6.5%, 7.1% and 7.6% respectively (S1 Table). According to the WHO criteria overweight and obesity prevalence at age three years was 5.2% and 1.8% respectively (S1 Table). At age seven years, the mean (SD) BF% was calculated at 19.1% (±5.1%) and 21.5% (±5.6%) for boys and girls respectively. The respective values at age fourteen years were 14.9% (±8.2%) and 26.6% (±7.0%).

Infants with missing data tended to have mothers that were younger, had General Certificate of Secondary Education grades D-G and an income of 0–10399 UK pounds–S2 Table.

The mean BMI by the four birth modes is depicted at each of the five time points, from age three to fourteen years, in S1 Fig. On average, mean BMI was lowest for normal VD and highest for planned CS. The mean BMI reached its nadir, of 16.3 kg/m$^2$ at age five years. Fig 1 depicts the mean BMI for all VD and CS births; it was highest for the latter. Those born by planned CS had a mean BMI that was similar to those born by normal VD (adjusted mean difference = 0.00; [95% confidence interval (CI) -0.10; 0.11], p-value = 0.97) (Table 2). For those born by emergency CS the adjusted mean difference was 0.08; [95% CI -0.01; 0.17], p-value = 0.09.

There was no association between planned CS and any BMI category transition, S3 Table. The adjusted relative risk ratio of remaining obese from the age of three to five years among those born by emergency CS was 1.34; [95% CI 0.98; 1.82], p-value = 0.07.

At age seven years, there was no association between planned CS and BF% (adjusted BF% mean difference = 0.13; [95% CI -0.23; 0.49], p-value = 0.47) and emergency CS (adjusted BF% mean difference = 0.21; [95% CI -0.11; 0.54], p-value = 0.20) in comparison to the reference group of children delivered by unassisted VD (Table 3). At age fourteen years, there was also no association (Table 3). Imputing missing maternal pre-pregnancy BMI and BF% did not alter our results materially (S4 Table). The prevalence of being overweight and obese in the observed data was almost identical to that of the pooled data.

Sub-group analysis for infants with mothers > 35 years old, born pre-term or by their sex did not reveal any statistically significant results (S5–S8 Tables).

## Discussion

### Main findings

From a large contemporary prospective longitudinal cohort study, we found that infants born by planned CS did not have an increased BMI overall, from age three to fourteen years, compared with those born by normal VD. We also found that obesity prevalence increased from age three years onwards. Infants born by planned CS did not have an increased BF% at age seven and fourteen years compared with those born by normal VD.

### Interpretation

Our results are identical to those of another study that used MCS data, albeit at age three years. [41] This cross-sectional study, which estimated overweight risk in childhood from predictors

**Table 1. Characteristics of the study population.**

| Characteristic | Overall n (%) | Normal vaginal delivery n (%) | Assisted vaginal delivery [a] n (%) | Planned Caesarean section n (%) | Emergency Caesarean section n (%) |
|---|---|---|---|---|---|
| N | 18,116 (100) | 12,567 (69.4) | 1677 (9.3) | 1669 (9.2) | 2203 (12.2) |
| Maternal age (years), median IQR | 29 (24–33) | 28 (23–32) | 29 (24–32) | 31 (27–34) | 30 (25–33) |
| < 20 | 1572 (8.7) | 1,214 (9.7) | 171 (10.2) | 42 (2.5) | 145 (6.6) |
| 20–24 | 3491 (19.3) | 2,643 (21.0) | 291 (17.4) | 207 (12.4) | 350 (15.9) |
| 25–29 | 5010 (27.7) | 3,491 (27.8) | 505 (30.1) | 409 (24.5) | 605 (27.5) |
| 30–34 | 5215 (28.8) | 3,447 (27.4) | 479 (28.6) | 605 (36.2) | 684 (31.0) |
| 35–39 | 2443 (13.5) | 1,541 (12.3) | 210 (12.5) | 342 (20.5) | 350 (15.9) |
| ≥ 40 | 382 (2.1) | 228 (1.8) | 21 (1.3) | 64 (3.8) | 69 (3.1) |
| Ethnicity | | | | | |
| European | 15,180 (83.3) | 10,411 (82.2) | 1,525 (90.9) | 1,426 (85.4) | 1,818 (82.5) |
| Asian | 1,911 (10.5) | 1,424 (11.3) | 101 (6.0) | 163 (9.8) | 223 (10.1) |
| African | 664 (3.7) | 464 (3.7) | 20 (1.2) | 51 (3.1) | 129 (5.9) |
| Mixed | 186 (1.0) | 134 (1.1) | 15 (0.9) | 17 (1.0) | 20 (0.9) |
| Any other background | 146 (0.8) | 107 (0.9) | 15 (0.9) | 11 (0.7) | 13 (0.6) |
| Missing | 29 (0.2) | 27 (0.2) | 1 (0.1) | 1 (0.1) | 0 (0.0) |
| Highest education | | | | | |
| GCSE grades D-G | 1,944 (10.7) | 1,392 (11.1) | 158 (9.4) | 163 (9.8) | 231 (10.5) |
| O level / GCSE grades A-C | 6,047 (33.4) | 4,202 (33.4) | 567 (33.8) | 570 (34.2) | 708 (32.1) |
| A / AS / S levels | 1,687 (9.3) | 1,153 (9.2) | 183 (10.9) | 137 (8.2) | 214 (9.7) |
| Diplomas in higher education | 1,511 (8.3) | 962 (7.7) | 179 (10.7) | 166 (9.9) | 204 (9.3) |
| First degree | 2,229 (12.3) | 1,369 (10.9) | 302 (18.0) | 218 (13.1) | 340 (15.4) |
| Higher degree | 604 (3.3) | 376 (3.0) | 66 (3.9) | 72 (4.3) | 90 (4.1) |
| Other academic qualifications (including overseas) | 526 (2.9) | 382 (3.0) | 37 (2.2) | 43 (2.6) | 64 (2.9) |
| None of these qualifications | 3,521 (19.4) | 2,691 (21.4) | 184 (11.0) | 299 (17.9) | 347 (15.8) |
| Missing | 47 (0.3) | 40 (0.3) | 1 (0.1) | 1 (0.1) | 5 (0.2) |
| Total net couple income (UK pounds) | | | | | |
| 0–10399 | 1,858 (10.3) | 1,360 (10.8) | 136 (8.1) | 151 (9.0) | 211 (9.6) |
| 10400–15599 | 2,522 (13.9) | 1,837 (14.6) | 201 (12.0) | 209 (12.5) | 275 (12.5) |
| 15600–19799 | 2,533 (14.0) | 1,762 (14.0) | 241 (14.4) | 226 (13.5) | 304 (13.8) |
| 20800–30199 | 3,185 (17.6) | 2,089 (16.6) | 336 (20.0) | 334 (20.0) | 426 (19.3) |
| 31200–80000+ | 3,198 (17.7) | 1,984 (15.8) | 385 (23.0) | 371 (22.2) | 458 (20.8) |
| Not applicable | 3,525 (19.5) | 2,639 (21.0) | 271 (16.2) | 227 (13.6) | 388 (17.6) |
| Don't know | 921 (5.1) | 652 (5.2) | 64 (3.8) | 110 (6.6) | 95 (4.3) |
| Refused | 374 (2.1) | 244 (1.9) | 43 (2.6) | 41 (2.5) | 46 (2.1) |
| Marital status | | | | | |
| Legally separated | 516 (2.8) | 392 (3.1) | 24 (1.4) | 39 (2.3) | 61 (2.8) |
| Married, 1st and only marriage | 10016 (55.3) | 6,741 (53.6) | 958 (57.1) | 1,073 (64.3) | 1,244 (56.5) |
| Remarried, 2nd or later marriage | 730 (4.0) | 484 (3.9) | 46 (2.7) | 98 (5.9) | 102 (4.6) |
| Single never married | 6100 (33.7) | 4,419 (35.2) | 594 (35.4) | 370 (22.2) | 717 (32.5) |
| Divorced | 719 (4.0) | 507 (4.0) | 53 (3.2) | 83 (5.0) | 76 (3.4) |
| Widowed | 33 (0.2) | 22 (0.2) | 2 (0.1) | 6 (0.4) | 3 (0.1) |
| Missing | 2 (0.0) | 2 (0.0) | 0 (0.0) | 0 (0.0) | 0 (0.0) |
| Body mass index ($kg/m^2$) pre-pregnancy, median IQR | 22.7 (20.6–25.7) | 22.5 (20.6–25.3) | 22.5 (20.7–25.1) | 23.7 (21.4–27.1) | 23.4 (21.2–26.8) |

*(Continued)*

**Table 1.** (Continued)

| Characteristic | Overall n (%) | Normal vaginal delivery n (%) | Assisted vaginal delivery [a] n (%) | Planned Caesarean section n (%) | Emergency Caesarean section n (%) |
|---|---|---|---|---|---|
| Missing | 1558 (8.6) | 1,110 (8.8) | 96 (5.7) | 159 (9.5) | 193 (8.8) |
| Smoking during pregnancy | | | | | |
| Non-smoker | 12,927 (71.4) | 8,935 (71.1) | 1,169 (69.7) | 1,244 (74.5) | 1,579 (71.7) |
| Gave up | 2,298 (12.7) | 1,526 (12.1) | 268 (16.0) | 208 (12.5) | 296 (13.4) |
| Smoker | 2,877 (15.9) | 2,094 (16.7) | 239 (14.3) | 216 (12.9) | 328 (14.9) |
| Missing | 14 (0.1) | 12 (0.1) | 1 (0.1) | 1 (0.1) | 0 (0.0) |
| Diabetes mellitus | | | | | |
| Any kind of diabetes mellitus | 313 (1.7) | 144 (1.1) | 18 (1.1) | 79 (4.7) | 72 (3.3) |
| No diabetes mellitus | 17,802 (98.3) | 12,422 (98.8) | 1,659 (98.9) | 1,590 (95.3) | 2,131 (96.7) |
| Missing | 1 (0.0) | 1 (0.0) | 0 (0.0) | 0 (0.0) | 0 (0.0) |
| Number of other children–'parity' | | | | | |
| 1 | 17,474 (96.5) | 12,113 (96.4) | 1,663 (99.2) | 1,571 (94.1) | 2,127 (96.6) |
| 2 | 470 (2.6) | 320 (2.5) | 11 (0.7) | 83 (5.0) | 56 (2.5) |
| 3+ | 168 (0.9) | 131 (1.0) | 3 (0.2) | 15 (0.9) | 19 (0.9) |
| Missing | 4 (0.0) | 3 (0.0) | 0 (0.0) | 0 (0.0) | 1 (0.0) |
| Sex | | | | | |
| Male | 9,322 (51.5) | 6,330 (50.4) | 930 (55.5) | 814 (48.8) | 1,248 (56.7) |
| Female | 8,794 (48.5) | 6,237 (49.6) | 747 (44.5) | 855 (51.2) | 955 (43.3) |
| Gestational age (weeks) | | | | | |
| Preterm (< 37) | 1708 (9.4) | 978 (7.8) | 100 (6.0) | 178 (10.7) | 452 (20.5) |
| Term (37–41) | 15,992 (88.3) | 11,306 (90.0) | 1,535 (91.5) | 1,467 (87.9) | 1,684 (76.4) |
| Postterm (> 42) | 225 (1.2) | 147 (1.2) | 28 (1.7) | 6 (0.4) | 44 (2.0) |
| Missing | 191 (1.1) | 136 (1.1) | 14 (0.8) | 18 (1.1) | 23 (1.0) |
| Birth weight (kg), median IQR | 3.37 (3.03–3.71) | 3.37 (3.04–3.71) | 3.43 (3.15–3.77) | 3.35 (3–3.69) | 3.36 (2.84–3.80) |
| Missing | 14 (0.1) | 11 (0.1) | 0 (0.0) | 3 (0.2) | 0 (0.0) |
| Macrosomia (> 4kg) | 1,957 (10.8) | 1,264 (10.1) | 184 (11.0) | 177 (10.6) | 332 (15.1) |

UK (United Kingdom), SD (Standard deviation), IQR (Interquartile range), GCSE (General Certificate of Secondary Education).

Vacuum or forceps [a]

during infancy, found no association between CS birth and being overweight at age three years. One of the few studies to utilise within family analysis, in addition to traditional observational cohort analytic techniques, also found no association between CS birth and childhood obesity.[42] The national representativeness and the generalisability of this MCS study result to the UK population is reinforced by similar CS rates of ~21% in this cohort and in the general population at the turn of the second millennium.[43]

As we previously reported using a different cohort, there was no association between planned/elective CS delivery and obesity or transition into or out of obesity between ages three and five years.[10]

The natural history of BMI across the life course identifies peak BMI during the first two years of life which then reaches the lowest post infancy values at around five years of age.[44] This takes into account that infants born by CS have a higher BMI than those born by VD. We too found this BMI pattern, namely a nadir around age five, and CS infants having a non-significantly higher BMI.[10, 31] Cross sectional analysis of the association between mode of

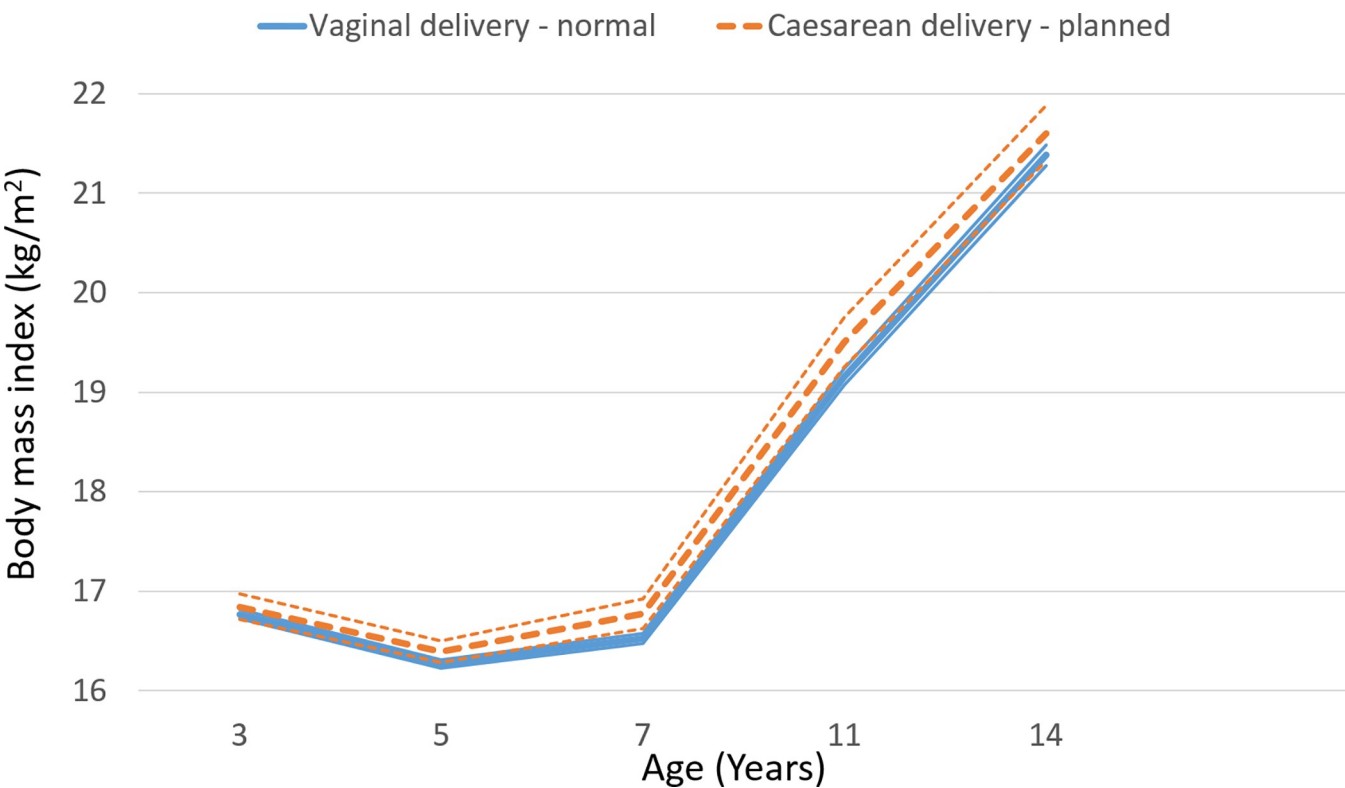

**Fig 1. Mean body mass index by birth mode from age three to fourteen years with 95% confidence intervals–thin lines–for non-macrosomic infants born by normal vaginal delivery and by planned Caesarean section.**

birth and BMI would therefore be influenced by the natural history and the age at which analysis was done. Therefore the first two years of life, during which BMI reaches a peak seems to be when the greatest, statistically significant, divergence in BMI between CS and VD born infants occurs.[14, 31, 44]

The prevalence of childhood obesity, in our study, did not follow a trajectory wherein it declines from age two to fourteen.[45] This may be due to the global childhood obesity epidemic driven by positive caloric intake.[29] In the MCS, family lifestyle may also have been contributory.[46]

That delivery mode is not associated with BF%, in both girls and boys, has been reported from a Brazilian longitudinal cohort study, and also in our previous publication.[31, 47]

**Table 2. Mode of birth and body mass index.**

| BMI | Coef (95% CI) | p-value | AdjCoef (95% CI)** | p-value |
|---|---|---|---|---|
| Normal vaginal | reference | | reference | |
| Assisted vaginal | -0.08 (-0.18; 0.02) | 0.116 | -0.03 (-0.13; 0.07) | 0.567 |
| Planned Caesarean | 0.18 (0.08; 0.28) | 0.000 | 0.00 (-0.10; 0.10) | 0.971 |
| Emergency Caesarean | 0.18 (0.09; 0.27) | 0.000 | 0.08 (-0.01; 0.17) | 0.091 |

Time points for adjusted model = 50,917 at ages three, five, seven, eleven and fourteen years. Mixed-effects linear regression. BMI–Body mass index, Coef (Coefficient), CI (Confidence intervals), Adj (Adjusted).

**Adjusted for maternal age, ethnicity, education, marital status, couple income, infant sex, birth weight, smoking, gestational age, diabetes mellitus, parity, pre-pregnancy BMI (Non-macrosomic infants).

**Table 3. Mode of delivery and body fat percent at seven and fourteen years.**

| Delivery mode (seven years) | Coef. (95% CI) | p-value | AdjCoef. (95% CI)** | p-value |
|---|---|---|---|---|
| Normal vaginal delivery | reference | | reference | |
| Assisted vaginal | -0.21 (-0.56; 0.14) | 0.248 | 0.03 (-0.31; 0.37) | 0.864 |
| Planned Caesarean | 0.43 (0.08; 0.78) | 0.016 | 0.13 (-0.23; 0.49) | 0.466 |
| Emergency Caesarean | 0.35 (0.03; 0.67) | 0.032 | 0.21 (-0.11; 0.54) | 0.199 |
| Delivery mode (fourteen years) | Coef. (95% CI) | p-value | AdjCoef. (95% CI)** | p-value |
| Normal vaginal delivery | reference | | reference | |
| Assisted vaginal | -1.26 (-1.91; -0.61) | 0.000 | -0.40 (-0.94; -0.13) | 0.139 |
| Planned Caesarean | 0.50 (-0.16; 1.15) | 0.135 | -0.08 (-0.64; 0.47) | 0.769 |
| Emergency Caesarean | -0.04 (-0.62; -0.55) | 0.904 | -0.00 (-0.50; 0.50) | 0.999 |

N for adjusted model = 10,254 and 8,279 at age seven and fourteen respectively. Linear regression. Coef (Coefficient), CI (Confidence intervals), Adj (Adjusted).

**Adjusted for maternal age, ethnicity, education, marital status, couple income, infant sex, birth weight, smoking, gestational age, diabetes mellitus, parity, pre-pregnancy body mass index (Non-macrosomic infants).

Disparate findings were reported from a Mexican study (n = 256) which also used bioelectric impedance to assess body composition at approximately age seven years.[48] Girls, but not boys, born by CS had a higher fat mass index although no distinction was made between planned and emergency CS. Our main findings are similar to those reported in adolescents, aged fifteen years, where, after adjusting for potential confounders, no association was found between CS birth and obesity—as defined according to WHO Standards.[49] A United States study, albeit with a sample size of less than a thousand, found that delivery type did not predict obesity in adolescence. [50] These aforementioned results would be in keeping with how the infant microbiota undergoes considerable reorganisation in the first six weeks of life which is influenced by body site rather than by delivery mode.[17] Disparate findings have been reported, with obesity rates higher in twenty year olds delivered by CS, although the underlying sample was not nationally representative, thereby reducing external validity.[13]. The exposures planned and emergency CS likely have different confounding structures. Although the results were null for both types of exposure, the point estimates were generally greater for emergency CS than for planned CS which is reflective of this underlying dissimilar confounding structure. Around the time of puberty,[51] an acceleration of BMI towards adult values was observed at age eleven and fourteen years, however the association between delivery mode and BMI remained non-significant.

## Strengths and limitations

Firstly, the MCS cohort is a large nationally representative prospective study which allows ready generalisation of findings to the population. In contemporary literature, the baseline sample size of over 18,000 represents one of the largest cohorts and the follow-up to age fourteen years is one of the longest thus far perfomed.[10, 14] Secondly, maternal pre-pregnancy BMI, a key confounder, was available, thus mitigating a key limitation of previous analyses.[3] Thirdly, it was possible to separate CS birth into planned and emergency CS which only a limited number of earlier studies have managed to do.[10–12, 14] Fourthly, having children born during every month of the year mitigated the effects of seasonality. This was important since birth month can be a proxy for seasonal attributes which may influence future health.[52]

With planned CS, membranes were unlikely to have ruptured as women were not in labour. Since our hypothesis was based on pre-labour CS, the classification of CS[53] into planned and emergency was unlikely to have influenced our results. Although the final mode of birth was obtained from mothers approximately nine months post-partum, maternal recall of delivery

mode in the MCS has been demonstrated to be reliable, (approximately 98% of mothers recalled this accurately).[54] Paucity of phenotypic data from fathers represents a constraint because they have been demonstrated to play a significant role in the development of childhood obesity.[55] We did not have data that permitted within family analysis.[8, 9] Due to unavailability of data on antibiotics administered intrapartum, our results were not adjusted for this potentially confounding factor. However, we are confident that this limitation did not alter our results because previous studies that adjusted for intrapartum antibiotic administration did not have their results changed materially.[14, 44] The confounding factor maternal gestational weight gain, which is linked to post-pregnancy weight retention, was not available. This limited our study. However because of the high degree of correlation between pre-pregnancy BMI and gestational weight gain we believe our models had sufficient merit.[56, 57] Using bioelectric impedance, for large studies like the MCS, is advantageous because of its portability, ease of use and low cost; the disadvantage however is that bioelectric impedance underestimates BF%.[58] Using other BMI classification, like the WHO system, would not change the results of the comparisons of the absolute values of BMI.

Most CS births are performed under regional anaesthesia, thus the kind of anaesthesia was unlikely to have contributed to our results.[59] It was not possible to rule out possible confounding due to the underlying reasons for CS because there were no further variables like previous CS available to capture the health of the mother prior to birth and the exact indications for CS birth were unavailable. In addition, as for any observational study, it was not possible to completely exclude residual confounding. Attrition of participants, which was more pronounced at later ages–up to 43.3%, also represents a limitation. Multiple imputation suggested that this missing data did not affect our results. Although there was inherent lack of power for some analyses, particularly at later ages because of loss to follow-up, consistency of the results suggests their merit.

## Conclusion

Infants born by planned CS did not have a significantly higher BMI or BF% compared to those born by normal VD. This may suggest that the association described in the literature could be due to the indications/reasons for CS birth or residual confounding.

## Supporting information

**S1 Table. International Obesity Task Force classification of body mass index from age three to fourteen and body fat% at age seven and fourteen.**
(PDF)

**S2 Table. Missing data for body mass index at age two years.**
(PDF)

**S3 Table. Mode of delivery and BMI category transition between ages three and five.**
(PDF)

**S4 Table. Mode of delivery and body fat percent at seven and fourteen years.** Imputed pre-pregnancy BMI and childhood body fat percent.
(PDF)

**S5 Table. Mode of birth and body mass index for infants with mothers > 35 years old.**
(PDF)

**S6 Table. Mode of birth and body mass index for infants born pre-term.**
(PDF)

**S7 Table. Mode of birth and body mass index for male infants.**
(PDF)

**S8 Table. Mode of birth and body mass index for female infants.**
(PDF)

**S1 Fig. Mean body mass index by birth mode from age three to fourteen years.**
(PDF)

## Acknowledgments

We acknowledge and thank the MCS participants.

## Author Contributions

**Conceptualization:** Gwinyai Masukume, Ali S. Khashan, Susan M. B. Morton, Philip N. Baker, Louise C. Kenny, Fergus P. McCarthy.

**Formal analysis:** Gwinyai Masukume.

**Funding acquisition:** Susan M. B. Morton, Philip N. Baker, Louise C. Kenny.

**Methodology:** Gwinyai Masukume, Ali S. Khashan, Susan M. B. Morton, Philip N. Baker, Louise C. Kenny, Fergus P. McCarthy.

**Supervision:** Ali S. Khashan, Susan M. B. Morton, Philip N. Baker, Louise C. Kenny, Fergus P. McCarthy.

**Writing – original draft:** Gwinyai Masukume.

**Writing – review & editing:** Ali S. Khashan, Susan M. B. Morton, Philip N. Baker, Louise C. Kenny, Fergus P. McCarthy.

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
