## [Decision Letter · Decision Letter 0]

28 Jul 2019

PONE-D-19-17591

Caesarean Section Delivery and Childhood Obesity in a British Longitudinal Cohort Study

PLOS ONE

Dear Dr Masukume,

Thank you for submitting your manuscript to PLOS ONE. After careful consideration, we feel that it has merit but does not fully meet PLOS ONE’s publication criteria as it currently stands. Therefore, we invite you to submit a revised version of the manuscript that addresses the points raised during the review process.

Please address all the Reviewers' remarks in your revised manuscript, with a special attention to the points made by Reviewer 1 regarding the definitions of weight categories, and the methodology of the study.

In addition, we feel that some important, more subtle markers of metabolic health in and around pregnancy are missing as potentially confounding factors, such as weight gain during pregnancy and post-pregnancy weight retention, and would suggest their addition, would they be available.  

We would appreciate receiving your revised manuscript by Sep 11 2019 11:59PM. To enhance the reproducibility of your results, we recommend that if applicable you deposit your laboratory protocols in protocols.io, where a protocol can be assigned its own identifier (DOI) such that it can be cited independently in the future. For instructions see: http://journals.plos.org/plosone/s/submission-guidelines#loc-laboratory-protocols

We look forward to receiving your revised manuscript.

Kind regards,

Umberto Simeoni

Academic Editor

PLOS ONE

Journal Requirements:

Reviewers' comments:

Reviewer's Responses to Questions

**Comments to the Author**

1. Is the manuscript technically sound, and do the data support the conclusions?

Reviewer #1: Yes

Reviewer #2: Yes

2. Has the statistical analysis been performed appropriately and rigorously? 

Reviewer #1: Yes

Reviewer #2: Yes

3. Have the authors made all data underlying the findings in their manuscript fully available?

Reviewer #1: Yes

Reviewer #2: Yes

4. Is the manuscript presented in an intelligible fashion and written in standard English?

Reviewer #1: Yes

Reviewer #2: Yes

5. Review Comments to the Author

Reviewer #1: This paper brings significant information about the impact of delivery on later risk of obesity. The cohort and statistical study are well designed.

However, the criteria used to define weight categories should be updated since there is a risk of misclassification:

The IOTF criteria rely on 6 populations sample around the world but do not represent the growth pattern of a real child. Nowadays, WHO reference curves are much better tools which can be used in both clinical and epidemiological settings. They should be used instead since they reflect actual growth patterns.

Another point is the use of bioimpedance in order to measure body composition: a recent study by Tanita reports a mean difference in body composition with DEXA of -6.75 % with what seems to be one of more recent device of this company. BIA is not considered a valid tool in paediatrics. This limit should be pinpointed. This should lead to the conclusion that BMI alone may be a sufficient tool in the field of epidemiology in paediatrics.

The impact of puberty on the older category (14 years) is not quoted at all, albeit certain.

Although using other BMI classification would not change the results of the comparisons of the absolute values of the BMI, it may bring different information about the shiftes form categories of BMI or the proportions of children in each category.

The strength of this paper would be very much improved using up to date paediatric tools.

Reviewer #2: This is a well written paper that clearly motivates the contribution it makes over the prior literature. It is also an important topic on which to publish null findings. The data they use is well suited for testing their hypothesis. They also have an excellent description of the proposed mechanisms. The analysis is well done, including multiple imputation for missing data, and I have only a few minor comments/suggestions.

1. On line 108, do the authors mean “accurate” rather than “precise”?

2. It would be useful in the introduction to discuss the estimates of association with obesity published in the literature, ideally from the meta-analyses, so the adequacy of power to detect a null hypothesis could be assessed. That is, is the sample size here enough to detect the non-null coefficients that are presumably biased?

3. The separation of planned and emergency CS is very innovative. I think the authors could highlight more specifically the fact that these two procedures likely have different confounding structures, thus the fact that results are null for both types of exposures adds additional credibility to their null findings.

4. Are there any further variables to capture the health of the mother prior to birth that could be controlled for? I think the list is adequate, but in particular curious about indicators of poor health that would predict emergency C-section. Also, given critical importance of maternal pre-pregnancy BMI as a confounder, how was this controlled for? It should be used in a way that accounts for non-linearity of confounding.

6. PLOS authors have the option to publish the peer review history of their article (what does this mean?). If published, this will include your full peer review and any attached files.

Reviewer #1: No

Reviewer #2: No

---

## [Author Response · Author response to Decision Letter 0]

18 Aug 2019

The Irish Centre for Fetal and Neonatal Translational Research (INFANT)

5th Floor Cork University Maternity Hospital

Wilton, Cork, Republic of Ireland

+353 830409737 

Fergus.mccarthy@ucc.ie

18th August 2019

The Editor

PLOS ONE

Dear Editor,

RE: PONE-D-19-17591 Caesarean Section Delivery and Childhood Obesity in a British Longitudinal Cohort Study

Thank you for sending our manuscript for review and for now considering it for publication. We are grateful to the peer reviewers for their helpful comments which strengthen our manuscript. We respond point-by-point to each comment in the attached response to reviewers - Microsoft Word file.

We hope that we have provided sufficient clarification. We look forward to your consideration.

Kindest regards,

Gwinyai Masukume

MB ChB, Dip Obst, MSc

On behalf of all Authors

---

## [Decision Letter · Decision Letter 1]

1 Oct 2019

Caesarean Section Delivery and Childhood Obesity in a British Longitudinal Cohort Study

PONE-D-19-17591R1

Dear Dr. Masukume,

We are pleased to inform you that your manuscript has been judged scientifically suitable for publication and will be formally accepted for publication once it complies with all outstanding technical requirements.

With kind regards,

Umberto Simeoni

Academic Editor

PLOS ONE

Additional Editor Comments (optional):

Reviewers' comments:

Reviewer's Responses to Questions

**Comments to the Author**

1. If the authors have adequately addressed your comments raised in a previous round of review and you feel that this manuscript is now acceptable for publication, you may indicate that here to bypass the “Comments to the Author” section, enter your conflict of interest statement in the “Confidential to Editor” section, and submit your "Accept" recommendation.

Reviewer #1: All comments have been addressed

Reviewer #2: All comments have been addressed

2. Is the manuscript technically sound, and do the data support the conclusions?

Reviewer #1: Yes

Reviewer #2: Yes

3. Has the statistical analysis been performed appropriately and rigorously? 

Reviewer #1: Yes

Reviewer #2: Yes

4. Have the authors made all data underlying the findings in their manuscript fully available?

Reviewer #1: Yes

Reviewer #2: Yes

5. Is the manuscript presented in an intelligible fashion and written in standard English?

Reviewer #1: Yes

Reviewer #2: Yes

6. Review Comments to the Author

Reviewer #1: This manuscript has been improved. It provides interesting data to scientific community.

We think that IOTF criteria should in the future be definitely ruled out because the WHO cohorts were adequatly designed and because IOTF remains a statistical tool only.

A last point has to be changed: the authors speculate about vaginal microflora and microbiota but do not bring any data, so these terms should not be quoted about the key words.

Reviewer #2: I have no further comments, thank you for your attention to the prior reviews. This manuscript makes an excellent contribution to the literature.

7. PLOS authors have the option to publish the peer review history of their article (what does this mean?). If published, this will include your full peer review and any attached files.

Reviewer #1: No

Reviewer #2: No

---

## [Editor Report · Acceptance letter]

9 Oct 2019

PONE-D-19-17591R1 

Caesarean Section Delivery and Childhood Obesity in a British Longitudinal Cohort Study 

Dear Dr. Masukume:

I am pleased to inform you that your manuscript has been deemed suitable for publication in PLOS ONE. Congratulations! Your manuscript is now with our production department. 

With kind regards,

on behalf of

Dr. Umberto Simeoni 

Academic Editor

PLOS ONE